# Post-Movement Beta Synchrony Inhibits Cortical Excitability

**DOI:** 10.3390/brainsci14100970

**Published:** 2024-09-26

**Authors:** Edward Rhodes, William Gaetz, Jonathan Marsden, Stephen D. Hall

**Affiliations:** 1Brain Research & Imaging Centre, University of Plymouth, Plymouth PL4 8AA, UK; e.rhodes@imperial.ac.uk (E.R.); jonathan.marsden@plymouth.ac.uk (J.M.); 2UK Dementia Research Institute, Imperial College London, London W1T 7NF, UK; 3Department of Radiology, Children’s Hospital of Philadelphia, Philadelphia, PA 19104, USA; 4School of Health Professions, University of Plymouth, Plymouth PL6 8BH, UK

**Keywords:** beta synchrony, movement, motor control, oscillations, PMBR, B-ERD, cortical excitability

## Abstract

Background/Objectives: This study investigates the relationship between movement-related beta synchrony and primary motor cortex (M1) excitability, focusing on the time-dependent inhibition of movement. Voluntary movement induces beta frequency (13–30 Hz) event-related desynchronisation (B-ERD) in M1, followed by post-movement beta rebound (PMBR). Although PMBR is linked to cortical inhibition, its temporal relationship with motor cortical excitability is unclear. This study aims to determine whether PMBR acts as a marker for post-movement inhibition by assessing motor-evoked potentials (MEPs) during distinct phases of the beta synchrony profile. Methods: Twenty-five right-handed participants (mean age: 24 years) were recruited. EMG data were recorded from the first dorsal interosseous muscle, and TMS was applied to the M1 motor hotspot to evoke MEPs. A reaction time task was used to elicit beta oscillations, with TMS delivered at participant-specific time points based on EEG-derived beta power envelopes. MEP amplitudes were compared across four phases: B-ERD, early PMBR, peak PMBR, and late PMBR. Results: Our findings demonstrate that MEP amplitude significantly increased during B-ERD compared to rest, indicating heightened cortical excitability. In contrast, MEPs recorded during peak PMBR were significantly reduced, suggesting cortical inhibition. While all three PMBR phases exhibited reduced cortical excitability, a trend toward amplitude-dependent inhibition was observed. Conclusions: This study confirms that PMBR is linked to reduced cortical excitability, validating its role as a marker of motor cortical inhibition. These results enhance the understanding of beta oscillations in motor control and suggest that further research on altered PMBR could be crucial for understanding neurological and psychiatric disorders.

## 1. Introduction

Voluntary movement in humans has been associated with well-established patterns of neural oscillatory activity within the primary motor cortex (M1). Prior to the onset, and during the execution of a movement, there is a reliable event-related desynchronisation in the beta (13–30 Hz) frequency range, typically termed ‘beta event-related desynchronisation’ (B-ERD). Immediately following the cessation of movement, resynchronisation of the beta band occurs, which exceeds the pre-movement baseline, typically termed ‘post-movement beta rebound’ (PMBR). While ERD in the beta band is a robust correlate of motor preparation and execution [1,2,3,4,5], PMBR is a dependable oscillatory feature that occurs following movement termination [6,7,8,9,10]. Despite the sequential occurrence of B-ERD and PMBR, evidence suggests that they are physiologically and functionally discrete processes. Anatomically, B-ERD and PMBR have spatially distinct distributions across the motor system. ERD is observed in the supplementary motor area (SMA), primary somatosensory cortex (S1), posterior parietal cortex (PPC), and cerebellum [11,12,13,14,15,16], while PMBR is observed in M1, the premotor cortex (PMC), SMA and the frontal association cortex [12,15,17,18].

Importantly, the functional significance of B-ERD and PMBR is still unclear, although several possible explanations have been posited. Pfurtscheller and colleagues originally suggested that underlying beta activity represents an ‘idling’ status of the motor cortex [19]. This hypothesis suggests that B-ERD reflects cortical excitation during the recruitment of motor networks, for specific movement execution, and PMBR is a re-establishment of the inactive cortex upon movement completion [8,20]. Further studies propose that spontaneous beta activity promotes tonic postural control in the absence of voluntary movement [21], with increased beta effectively reinforcing the existing motor state to stabilise motor output [22,23,24]. A suggested consequence of this is the inhibition or impairment of new movement initiation [25,26]. A more fundamental association is also proposed, whereby PMBR facilitates the processing of movement-related sensory afference [9,27]. However, more recent studies demonstrate that PMBR indexes the processing of movement outcome [28,29] and the updating of pre-movement predictions [30]. For a recent review on the role of sensorimotor beta oscillations, see Barone and Rossiter [31].

In a ‘Go/NoGo’ task, the absence of movement following the ‘NoGo’ cue is accompanied by augmented beta synchrony [32]. The suggestion that this reflects a physiological inhibition of movement initiation is supported by the observation of lower amplitude motor evoked potentials (MEPs) following ‘NoGo’ compared to ‘Go’ cues [33], suggesting reduced motor cortical excitability. The inhibitory role of motoric beta synchronisation is further supported by impaired motor performance during M1 stimulation transcranial alternating current stimulation (tACS) at beta frequency [34]. 

Previous studies exploring the relationship between spontaneous beta activity and cortical excitability using MEP amplitude have reported varied results. While some studies find no relationship between beta and MEP amplitude [35,36], others report a negative correlation between beta and MEP [37]. Critically, TMS-evoked MEP studies demonstrate that stimulation within the putative PMBR window is associated with a reduced excitability of M1 neurons [33,38,39].

However, it is important to note that previous studies do not address the typically protracted time course of PMBR and the between-participant variance in the PMBR time-frequency signature. Specifically, the maximal PMBR amplitude is typically observed in a window 500–1000 ms after the completion of a movement and remains elevated for 1000–4000 ms [10,12,18,40]. While the within-participant retest reliability of the PMBR signal is robust for a specific movement, there is considerable between-participant variation [41]. Therefore, the accurate exploration of the inhibitory properties of the movement-related beta signal requires characterisation of the individual time-frequency envelope of beta power alongside measures of cortical excitability.

Here, we apply functionally targeted EEG to determine the participant-specific time-frequency envelope of movement-related beta modulation. This individual beta synchrony envelope was used to determine the temporal profile of TMS-evoked MEP amplitude as a marker of motor cortical excitability. The expectation in the present study is that beta synchrony is inversely proportional to the level of cortical excitability in the motor cortex. This prediction would result in minimal MEP amplitude during intervals of maximal PMBR and maximal MEP amplitude during intervals of minimum beta ERD. Accordingly, we hypothesised that: (1) the peak-to-peak amplitude of MEP, evoked at the peak PMBR period, will be lower than during all other phases of the motor-beta signature, and (2) the peak-to-peak amplitude of MEP, evoked at minimal beta power (immediately following cessation of movement), will be greater than all other phases of the motor-beta signature.

## 2. Methods

Twenty-five right-handed subjects were recruited (17 male), with a mean age of 24 years (SD = 12.62, range 18–69). Informed consent was obtained, and all studies were approved by the University of Plymouth Faculty Research Ethics Committee, in accordance with the ethical standards set by the 1964 Declaration of Helsinki. All subjects were screened using a TMS safety screening questionnaire, were free of medication, and did not have any personal or family history of neurological or psychiatric illness. Subject handedness was determined using the Edinburgh Handedness Inventory [42]. Three subjects were excluded from the analysis because of incomplete task performance.

### 2.1. Data Acquisition

#### 2.1.1. Electromyography Recording

Surface EMG activity was recorded from right first dorsal interosseous (FDI) muscles using a Bagnoli 2-channel hand-held EMG system (DelSys Inc., Boston, MA, USA), with reference ground electrodes placed over the ulnar process (Figure 1B). Two single differential surface electrodes, comprising two 10 mm silver bar strips spaced 10 mm apart, recorded muscle activity at 2048 Hz and digitised using a digital-analogue-converter (Power 1401-3, CED, Cambridge, UK). Data were recorded using Signal (Ver 6.4, CED, UK) with a 20 Hz–450 kHz bandwidth, 92 dB common-mode rejection ratio, and <1 kΩ input impedance.

#### 2.1.2. TMS Localisation of M1 and MEP Recording

Single-pulse TMS was performed using a Magstim 2002 stimulator, with a 70 mm diameter figure-of-eight coil (Magstim, Whitland, UK), held tangentially to the scalp with the coil handle pointing backwards 45° laterally [43,44]. The optimal position for evoking a response in the FDI muscle was marked on the scalp and the coil position was then fixed using a mechanical arm (Manfrotto & Co., Cassola, Italy). Resting motor threshold (RMT) was determined from the minimum stimulator output necessary to evoke an FDI MEP response with a peak-to-peak amplitude of at least 50 µV in 8 of 10 consecutive trials. Stimulator intensity +1% was defined as RMT for this experiment [45,46]. This ‘motor hotspot’ and stimulus amplitude was used throughout the experiment for the positioning of the EEG electrodes and subsequent generation of MEPs during the motor task. During the motor task, stimuli were delivered at 100% RMT.

#### 2.1.3. Electroencephalography (EEG) Recording

All EEG data were recorded using a DC-EEG feedback system (NEURO PRAX, neuroConn Germany). Ag/AgCl electrodes were arranged in a 5-electrode array (Figure 1D), centred on M1, following a previous protocol [47]. In brief, a central electrode was placed at the M1 motor hotspot, with four additional electrodes placed at 2 cm anterior, posterior, ventral and dorsal to the central electrode, to confirm the optimal position of the M1 electrode. EEG was referenced online to the ipsilateral mastoid and sampled at a rate of 2048 Hz, with impedance for all channels maintained below 10 kΩ. EEG signals were band-pass filtered with a bidirectional 2nd-order Butterworth band-pass filter between 1 and 35 Hz. EMG was recorded continuously to support movement detection and accurately time-lock TMS delivery to beta ERD and PMBR. During the MEP stage, EMG was used to measure the peak-to-peak amplitude of the induced MEPs.

### 2.2. Task Design

#### 2.2.1. Reaction Time Task

The same simple reaction time task was used for both EEG and TMS-MEP experimental phases (Figure 1A). Participants were seated comfortably, with stimuli displayed on a 21-inch, high-definition monitor, placed at a comfortable height, at ~90 cm in front of the participant. Subjects were instructed to focus on the centre of the presentation screen, with flat palms downward and with the right index finger resting on a force sensor (Mini S-Beam, Applied Measurements Ltd., UK), maintaining contact throughout the experiment (Figure 1B).

Each trial was 10 s in duration and began with a 1–1.5 s ‘wait’ phase, denoted by a red dot presented in the centre of the screen (Figure 1A). The timing of this phase was randomised on a trial-by-trial basis to reduce anticipation effects. A change from a red to a blue dot was the ‘Go’ cue, to which participants were instructed to ‘respond as quickly and accurately as possible with a downward press on the force sensor’. The cue remained on screen for 3 s, before being replaced by the red dot for a further randomised 5–5.5 s ‘rest’ period. This rest period ensured a period of 9–10 s between consecutive ‘go’ cues to ensure a reliable return to baseline following PMBR. The task was practiced prior to the experiment to ensure that participants learned to perform discrete sensor depression and relaxation, and to avoid additional finger-lift movements. Data were synchronised across experiments by the transmission of TTL pulses from the stimulus computer to the EEG and EMG recording systems at each transition point in the experiment.

#### 2.2.2. Experiment 1: EEG

EEG data were recorded continuously throughout Experiment 1. Participants performed the simple reaction-time task (described above), with stimulus time-locked markers used for further analyses. An initial 10-trial practice block was followed by four blocks of 40 trials, providing a total of 160 ten-second trials for analysis. Following completion, the EEG electrodes were removed from the scalp, while an automated analysis routine (see Section 2.3) computed the time-frequency envelope of the beta-frequency response to generate the stimulus parameters for Experiment 2 (Figure 1C).

### 2.3. Analysis

#### 2.3.1. EEG Analysis: Computing the Beta Power Envelope

EEG data were analysed offline using Matlab (Mathworks, Portola Valley, CA, USA) and FieldTrip [48]. Specifically, data from the recording montage were bandpass-filtered between 2 Hz and 100 Hz using a Hamming window-synced FIR filter and notch-filtered at 50 Hz to reduce electrical mains noise. Individual trials were time-locked, with zero defined as the onset of the ‘Go’ cue. Analysis trial epochs were 8.5 s in duration, beginning with a 500 ms baseline prior to cue onset (Figure 2). Initial analysis applied a Morlet wavelet time-frequency continuous wavelet transform (CWT) using the MATLAB Wavelet Toolbox. A fixed cycle width (w0 = 6) was used, with centre frequencies scaled across the 0–100 Hz range in 0.5 Hz increments, with a sampling rate of 2048 Hz. The power profile in the beta (13–30 Hz) range was computed, and the individual beta envelope was defined as the mean power at the beta range peak frequency ±2.5 Hz during the rest period. The change in mean beta power over each trial was computed for each participant, using a sliding window (10 samples), and normalised as a percentage of the mean power during the 500 ms baseline prior to cue-onset. From this, five specific time-points were defined for each participant for TMS stimulation in Experiment 2 (summarised in Table 1).

Force sensor and EMG data were used to identify trials in which no response occurred, or RT was outside the acceptable range (median RT ± 2*MAD). Trials with a poor EEG signal-to-noise ratio were also excluded from the analysis. A threshold of >20% trial removal was defined for participant exclusion; however, no participant exceeded this level. 

#### 2.3.2. Experiment 2: TMS-Evoked MEP

Following a 15-min break, participants were seated comfortably in a stereotactic frame, with the TMS coil positioned centrally over the M1 motor hotspot. Participants completed 200 trials of the same simple reaction time task, as described in Experiment 1 above (Figure 1C). The experiment was arranged into four blocks of 50 trials, each lasting 10 s. During each trial, the participant received one stimulation during the active rest period and a second at one of the four beta-change time points (Table 1 and Figure 3). MEPs were generated at the RMT as defined during the M1 localisation.

EMG was recorded continuously from the right FDI muscle during Experiment 2 (see EMG recording). EMG magnitude for each stimulation was defined as the peak-to-peak amplitude of the EMG signal following stimulation, and latency was defined as the interval from stimulation to peak EMG. For each participant, this generated 50 data points for each beta time-point, for comparison against 200 resting data points. Trials in which no MEP was induced were excluded from the analysis, as were any trials that contained outliers (median amplitude ± 2*MAD). For each subject, the average peak-to-peak amplitude of MEPs induced during the rest period was used as the baseline for normalisation.

To ensure that no MEP was generated during a period of muscle activation, triggers were not sent until EMG activity had fallen to within 0.5 SD of the pre-movement baseline mean.

#### 2.3.3. MEP Analysis

The effects of stimulation time-point on MEP amplitude were analysed with analysis of variance (ANOVA) with repeated measures. Time-point was used as the repeated measure and MEP amplitude as the dependent variable. Significant effects were further analysed using one-sample *t*-tests to test for a significant difference from the baseline (active rest period) and each of the four beta-change time-points. Further paired *t*-tests were then used to carry out planned contrasts between each of the four beta-related time-points to investigate changes in MEP amplitude.

## 3. Results

### 3.1. Experiment 1: EEG

Analysis of the time-frequency profile confirmed a characteristic pattern of beta ERD and PMBR in all participants (Figure 2). Analysis of the resting beta signal showed a prominent peak in the beta (13–30 Hz) range in all participants (mean 21.44 Hz, SD 3.787). Discrete time-points for TMS stimulation were obtained for all participants relative to movement and EMG activity termination (summarised in Table 1).

### 3.2. Experiment 2: TMS-Evoked EMG

Analysis confirmed a significant main effect of time-point on MEP amplitude (F(4,105) = 10.589, *p* < 0.0001, η2 = 0.287). To determine the cortical excitatory/inhibitory correspondence to phases of the beta synchrony profile, analysis of change in MEP amplitude compared to rest was conducted. This revealed that responses evoked during the ERD period immediately following movement (response termination) were significantly higher than at rest (mean difference (MD) = +26.86%, SD = 35.16%, t(21) = 3.582, *p* = 0.0018, d = 0.764). In contrast, MEPs induced during the peak PMBR period were significantly reduced in amplitude compared to rest (MD = −9.07%, SD = 18.66, t(21) = −2.28, *p* = 0.033, d = −0.486). Both the early (MD = −9.39%, SD = 21.9, t(21) = −2.009, *p* = 0.058, d = −0.429) and the late (MD = −5.45%, SD = 17.99, t(21) = −1.421, *p* = 0.17, d = −0.303) PMBR periods were also reduced in amplitude compared to rest, but not significantly (Figure 4).

To determine the extent to which the sub-phases of ERD and PMBR correspond to cortical excitation/inhibition, analysis of MEP amplitudes, between active periods, was conducted. This confirmed significant differences between ERD (response termination) and all three PMBR periods. ERD vs. early PMBR (t(21) = 4.027, *p* < 0.0001, d = 0.859), ERD vs. peak PMBR (t(21) = 4.177, *p* < 0.0001, d = 0.891), and ERD vs. late PMBR (t(21) = 3.939, *p* < 0.0001, d = 0.838). However, there was no significant difference in the elicited MEP amplitude between any of the three PMBR periods; early vs. peak PMBR (t(21) = 0.077, *p* = 0.939, d = 0.017), early vs. late PMBR (t(21) = −0.734, *p* = 0.471, d = −0.156) and peak vs. late PMBR (t(21) = −0.837, *p* = 0.412, d = −0.178) (Figure 4).

## 4. Discussion

As predicted, MEPs collected during the ERD period were significantly larger in amplitude than those collected at rest, while MEPs collected during the PMBR period showed a significant reduction in amplitude. These findings suggest that there is an inverse relationship between motor cortical beta power and cortical excitability, and that PMBR is a neural marker of post-movement cortical inhibition. Previous research has inferred a relationship between motor-related changes in motor cortex beta power and cortical excitability, based upon indirect measures. EEG/MEG studies have correlated MEP amplitudes with coincident power of spontaneous beta; in the absence of any movement [37,49] TMS-evoked MEP studies have previously characterised the time course of MEP amplitudes following completion of movement, without concurrent measures of underlying beta power [33,38,39]. These studies indicate a relationship between motoric beta power and motor cortex excitability. Here, we contribute a direct confirmation of this prominent theory through an EEG-directed TMS-evoked MEP study and confirm the following: that (i) cortical excitability is greater, as reflected by MEP amplitude, during a period of beta ERD [50], and that (ii) PMBR does indeed reflect post-movement cortical inhibition of the motor system [6,8,9,21,25,28]. We further demonstrate, through the testing of three discrete PMBR time-points (early, peak, and late), that (i) cortical inhibition persists for the duration of the PMBR signal, rather than following a time course fixed to either movement or initial beta deflection, and that (ii) while EMG differences between PMBR time-points were non-significant, the trend between strength of cortical inhibition and PMBR amplitude suggests an amplitude-dependent, rather than all-or-nothing, response. We suggest that further research could improve upon the present study through the continuous online measurement of EEG oscillatory activity alongside the delivery of TMS stimuli. Moreover, as previous research suggests, the oscillatory state in the experimental baseline is valuable for predicting subsequent motor actions [51,52] and evolves over the task duration [53]. Therefore, future studies should consider the influence of baseline state in modulating the identified inhibitory processes.

These findings are consistent with previous pharmacological observations that spontaneous and movement-related change in the sensorimotor beta signature is driven GABAergically [54,55,56,57,58] through spike-timing of inhibitory GABAergic interneurons [59]. Accordingly, pharmacological interventions demonstrate that the enhancement of GABAergic activity results in a reduction in MEP amplitude [60,61,62,63,64] and that motor impairment in movement disorders can be alleviated by a reduction in beta power [24]. 

Our results confirm the functional significance of PMBR as a signature of motor cortical inhibition. Future studies would benefit from the pairing of EEG/MEG with TMS to further detail the significance of altered PMBR in clinical populations, such as Multiple Sclerosis, Autistic Spectrum Disorder, and Parkinson’s disease [58,65,66], to further elucidate the functional significance of PMBR and, relatedly, the behavioural correlates associated with atypical PMBR signalling.

## 5. Conclusions

This study clarifies the functional implications of the PMBR signal that follows the execution of cued movement. Specifically, it affords an interval of reduced motor cortical excitability during which the threshold for initiation of a related motor function is increased. We suggest that these observations are consistent with the need for such an interval during post-movement periods to facilitate the information processing required to optimize future motor programmes.

## Figures and Tables

**Figure 1 brainsci-14-00970-f001:**
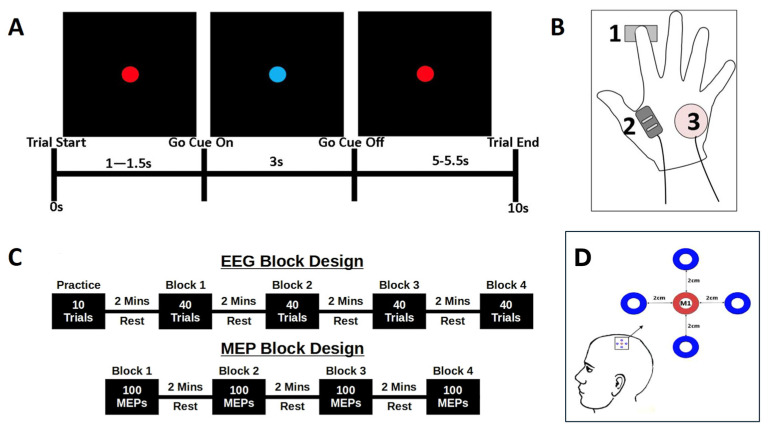
Experimental Protocol. (**A**) Schematic summary of a trial, showing the screen presentation and time course of the cue onset and duration, rest period, and overall trial length. (**B**) EMG measurement and response–device arrangement, showing the location of the force sensor beneath the tip of the index finger (1), locations of the FDI EMG sensor (2) and ulnar process reference (3). (**C**) EEG and MEP block designs, showing the arrangement of data acquisition in Experiments 1 (top) and 2 (bottom). (**D**) The 5 electrode EEG array centred on the functionally–localized M1.

**Figure 2 brainsci-14-00970-f002:**
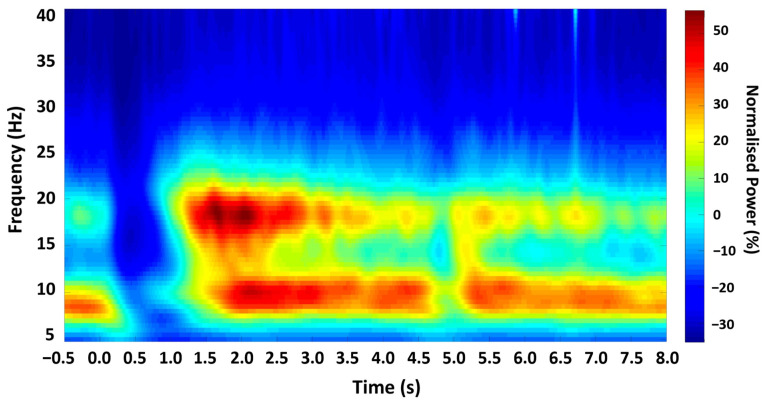
Time-frequency spectrogram. Grand-averaged Morlet Wavelet time-frequency analysis output from all participants, showing the mean oscillatory power in M1 across the experimental trial, with zero time-locked to cue onset.

**Figure 3 brainsci-14-00970-f003:**
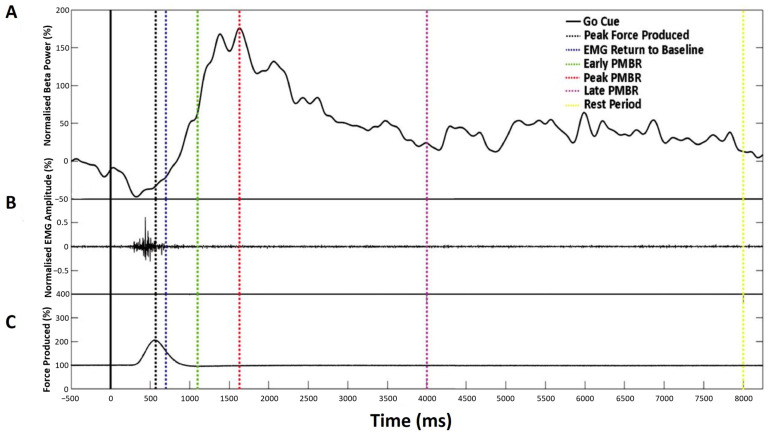
Characterisation of the beta-change time-points for TMS stimulation showing a representative single trial for an individual participant. (**A**) Normalised beta power at individual peak frequency, (**B**) Normalised EMG amplitude, and (**C**) Mean force production. Dashed vertical lines indicate the time-point selected for stimulation, as summarised in Table 1.

**Figure 4 brainsci-14-00970-f004:**
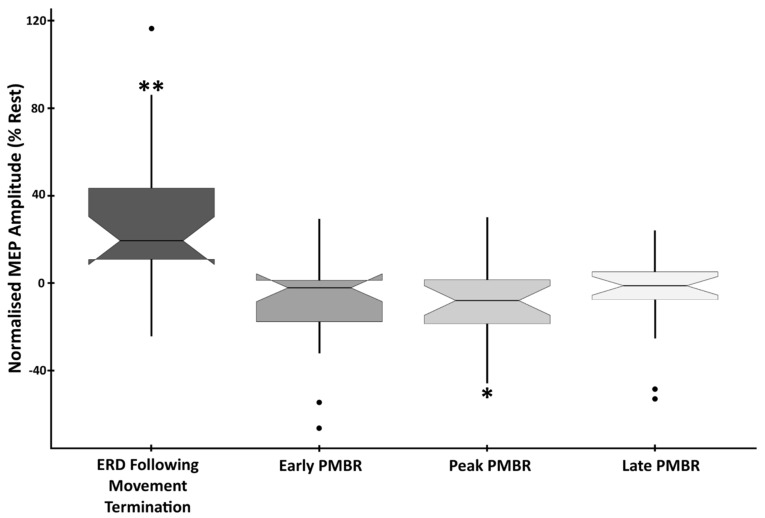
EMG Amplitude at beta-change time-points showing the difference from baseline in the mean peak-to-peak amplitude of MEPs induced during the four beta-change time-points. Data are normalised to the rest period, with statistically significant differences denoted as follows: ** *p* < 0.01; * *p* < 0.05.

**Table 1 brainsci-14-00970-t001:** Summary of the beta-change time-points across the participant cohort (*n* = 22), showing mean (SD) latencies and definitions of selection.

Stimulation Point	Time-Point (ms): Mean (SD)	Definition
Response termination	565.7 (±66.5)	During beta ERD, following response termination and after EMG activity returns to within 0.5 SD of baseline.
Early PMBR	1025.9 (±264.1)	Ascending slope at half the maximal amplitude of the PMBR.
Peak PMBR	1476.8 (±445.5)	Time-point of maximal PMBR amplitude.
Late PMBR	4186.4 (±674.9)	Descending slope immediately prior to return to mean baseline.
Active rest period	9500 (0)	500 ms prior to the end of each trial.

## Data Availability

The raw data supporting the conclusions of this article will be made available by the authors on request.

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
