# Peer review of "Post-Movement Beta Synchrony Inhibits Cortical Excitability"

_brainsci, 2024, doi:10.3390/brainsci14100970_

Round 1

Reviewer 1 Report

Comments and Suggestions for Authors

The manuscript is very interesting from a fundamental point of view, and the results confirm the functional significance of PMBR as a signature of motor cortical inhibition. 

Indeed, this study opens new perspectives in understanding the pathophysiology of psychomotor disorders.

However, the conclusions would have been more convincing if the authors had specified why they used a standardised rather than individually adopted approach to defining beta band limits? The second issue relates to the presence of muscle EMG artefacts in the EEG signal due to scalp psycho-emotional tension. This EMG signal is most coherent to the EEG signal just in the beta range. In other words, it would be good to provide evidence that PMBR is an EEG phenomenon and not scalp EMG.

Finally, when including women as subjects, it is always necessary to take into account their hormonal state (phase of the menstrual cycle) and, consequently, steroid hormone level-dependent psycho-emotional tension. At least in the discussion it would be good to clarify this factor, which may influence the results obtained. 

Author Response

The manuscript is very interesting from a fundamental point of view, and the results confirm the functional significance of PMBR as a signature of motor cortical inhibition. 

Indeed, this study opens new perspectives in understanding the pathophysiology of psychomotor disorders.

We thank reviewer 1 for their careful consideration and positive comments on the study.

However, the conclusions would have been more convincing if the authors had specified why they used a standardised rather than individually adopted approach to defining beta band limits?

The approach adopted in our study is exactly as reviewer 2 suggests, we have revised the manuscript to make this clearer. In summary, the initial use of a standardised (13-30Hz) band, was for the purpose of determining the peak beta power and confirming the expected ERD-PMBR profile. Individual beta was defined as peak beta +- 2.5 Hz. This process has now been clarified (lines 187-189).

The second issue relates to the presence of muscle EMG artefacts in the EEG signal due to scalp psycho-emotional tension. This EMG signal is most coherent to the EEG signal just in the beta range. In other words, it would be good to provide evidence that PMBR is an EEG phenomenon and not scalp EMG.

As described in the introduction, PMBR is a long-established phenomenon that has been localised to the sensorimotor cortices (lines 44-47). For example, the paper by Jurkiewicz [Reference 10] using source level analysis of MEG data (not susceptible to scalp muscle artifacts) confirms peak PMBR to occur in M1.

Finally, when including women as subjects, it is always necessary to take into account their hormonal state (phase of the menstrual cycle) and, consequently, steroid hormone level-dependent psycho-emotional tension. At least in the discussion it would be good to clarify this factor, which may influence the results obtained. 

While we agree that it is important to consider cyclic variation in factors influencing neural responses. We observed no difference between male and female participants and believe that including this point may introduce confusion for the reader.

Reviewer 2 Report

Comments and Suggestions for Authors

The authors investigate association between beta wave synchronicity and the primary motor cortex excitability. The paper looks interesting, and can be published subject to following revisions.

- The readability of Abstract could be improved, especially how the problem being addressed is introduced. Instead of Conclusions, better to refer to Findings (or Results or similar). Also 'enhance our understanding' is a bit vague, summarizing more specific findings would be desirable.

- It is always helpful to outline specific contributions (explicitly) in Introduction. Also, please add the last paragraph in Introduction outlining the structure of  the paper.

- Designating subsection without numbering is confusing. Please use subsection numbering. Could subsection labels on l. 112 and 146 deleted? Experiments could be in a separate subsection.

- In Discussion, please add limitations and any assumptions of your study. It helps to briefly outline future work.

Author Response

The authors investigate association between beta wave synchronicity and the primary motor cortex excitability. The paper looks interesting and can be published subject to following revisions.

We thank reviewer 2 for taking the time to review our manuscript and for the positive comments.

- The readability of Abstract could be improved, especially how the problem being addressed is introduced. Instead of Conclusions, better to refer to Findings (or Results or similar). Also 'enhance our understanding' is a bit vague, summarizing more specific findings would be desirable.

We have followed the required convention of the journal, in which the subheading ‘Conclusions’ is a requirement (https://www.mdpi.com/journal/brainsci/instructions#preparation). However, we have referred specifically to the study ‘findings’ in the results section (line 24) and attempted to clarify the formatting of the abstract section to make the subsections clear.

- It is always helpful to outline specific contributions (explicitly) in Introduction. Also, please add the last paragraph in Introduction outlining the structure of the paper.

We have sought to introduce the current state of understanding in this subject area, based upon an extensive range of contributions from past research. If reviewer 2 believes that specific methodology or observations requires further elaboration, we will be happy to address them specifically.

- Designating subsection without numbering is confusing. Please use subsection numbering. Could subsection labels on l. 112 and 146 deleted? Experiments could be in a separate subsection.

We have added subsection numbering for clarity and will be happy to amend subsections if the editors believe it will further clarify.

- In Discussion, please add limitations and any assumptions of your study. It helps to briefly outline future work.

We have suggested that further research could adopt an integrated TMS-EEG approach to further explore the observations described in our study (line 291-297).

Reviewer 3 Report

Comments and Suggestions for Authors

This study investigated the behavior of post-movement beta synchrony over a long duration, revealing that post-movement beta synchrony inhibits cortical excitability. The theme is interesting and has potential for significant contributions. However, several areas of the manuscript need improvement.

First, the introduction is somewhat incoherent, with relevant pieces of information scattered throughout. For instance, the statement "it is important to note that previous studies do not address the typically protracted time course of PMBR and the between-participant variance in the PMBR time-frequency signature" feels disconnected and abrupt. Additionally, the hypotheses should be derived with a stronger logical background, which is currently lacking. I believe the logic within the introduction should be clearly established.

There are also some technical details missing. For instance, how was signal synchronization achieved? Given that precise timing is essential for this analysis, synchronization needs to be validated before the experiment, but this information is not provided.

The details of EEG preprocessing are also absent. EEG signals are highly sensitive, and simply mentioning that an "automated analysis routine" was performed is insufficient. Since it appears the authors used FieldTrip, I assume standard preprocessing steps were followed. However, all these steps, along with the relevant parameters, must be fully disclosed. Typically, EEG preprocessing is detailed in a dedicated section with comprehensive descriptions.

I think the EEG channel locations can be presented in a figure for clarity.

The wavelet parameters used in the analysis are missing and should be provided.

How long was the inter-trial duration?

I also have concerns about the selection of the baseline. Given the varying period between the ‘trial start’ and the ‘go cue,’ it is likely that the baseline fluctuated across trials. There are several EEG studies that investigated the ‘premovement’ period [1,2,3], particularly the study [1] which divided the premovement period into subperiods, implying that a constant baseline during a varying premovement period might capture different motor planning phases. They found something during the period that you used as a baseline, which means their EEG signals were not still. I don’t think you can’t just ignore the previous studies. I suggest either performing an additional analysis to compare the selected baseline with the ‘pre-trial start’ period and (or at least) discussing this issue in the discussion section not by overlooking the importance of the ‘premovement’ period as demonstrated in the previous studies. I believe addressing this would strengthen the discussion 

[1] https://doi.org/10.3389/fnhum.2019.00063

[2] https://doi.org/10.3389/fneng.2012.00013

[3] https://doi.org/10.3389/fnins.2019.01148

Comments on the Quality of English Language

 The paper needs proofreading (e.g., “A suggested consequence being the inhibition or impairment of new movement initiation”)

Author Response

- This study investigated the behavior of post-movement beta synchrony over a long duration, revealing that post-movement beta synchrony inhibits cortical excitability. The theme is interesting and has potential for significant contributions. However, several areas of the manuscript need improvement.

We thank reviewer 3 for taking the time to carefully consider our manuscript, for the helpful suggestions and the positive comments on the value.

- First, the introduction is somewhat incoherent, with relevant pieces of information scattered throughout. For instance, the statement "it is important to note that previous studies do not address the typically protracted time course of PMBR and the between-participant variance in the PMBR time-frequency signature" feels disconnected and abrupt.

The subsequent section (see below) details the specifics of that introductory statement provide a clear explanation point: (lines 79-85).

“Specifically, the maximal PMBR amplitude is typically observed in a window 500-1000ms after the completion of a movement and remains elevated for 1000-4000ms [10,12,18,41]. While the within-participant retest reliability of the PMBR signal is robust for a specific movement, there is considerable between-participant variation [42]. Therefore, the accurate exploration of the inhibitory properties of the movement-related beta signal requires characterisation of the individual time-frequency envelope of beta power alongside measures of cortical excitability.”

If there is ambiguity about this or other statements in the introduction, we would welcome specific detail to enable us to clarify or expand.

- Additionally, the hypotheses should be derived with a stronger logical background, which is currently lacking. I believe the logic within the introduction should be clearly established. 

We have expanded the detail in the final section of the introduction to clarify the rationale for the hypothesis (lines 89-97).

- There are also some technical details missing. For instance, how was signal synchronization achieved? Given that precise timing is essential for this analysis, synchronization needs to be validated before the experiment, but this information is not provided.

These details have now been added (lines 160-162).

- The details of EEG preprocessing are also absent. EEG signals are highly sensitive, and simply mentioning that an "automated analysis routine" was performed is insufficient. Since it appears the authors used FieldTrip, I assume standard preprocessing steps were followed. However, all these steps, along with the relevant parameters, must be fully disclosed. Typically, EEG preprocessing is detailed in a dedicated section with comprehensive descriptions.

This detail has now been added (lines 179-183).

- I think the EEG channel locations can be presented in a figure for clarity. 

We have added a diagram showing the EEG montage to Figure 1D and referred to this in the text (line 131).

- The wavelet parameters used in the analysis are missing and should be provided.

These details have been added (lines 184-187)

- How long was the inter-trial duration? 

The period between the offset of the go cue and the start of the following trial was a randomized period of 5-5.5seconds. This allowed 9-10 seconds between consecutive ‘go’ cues to ensure that beta returned to baseline between trials. This is described, and has been elaborated in lines 156-158, and is shown in figure 1A.

- I also have concerns about the selection of the baseline. Given the varying period between the ‘trial start’ and the ‘go cue,’ it is likely that the baseline fluctuated across trials. I suggest either performing an additional analysis to compare the selected baseline with the ‘pre-trial start’ period and (or at least) discussing this issue in the discussion section not by overlooking the importance of the ‘premovement’ period as demonstrated in the previous studies. I believe addressing this would strengthen the discussion 

[1] https://doi.org/10.3389/fnhum.2019.00063

[2] https://doi.org/10.3389/fneng.2012.00013

[3] https://doi.org/10.3389/fnins.2019.01148

In our study, the beta power is computed on an individual trial basis, which minimizes the influence of fluctuation across the experimental period. Therefore, we do not believe that further analysis will be of value in the current study.

However, we agree that further research would benefit from characterizing the influence of baseline fluctuations to better understand how such modulation influences the evolution of inhibitory process in such studies. We have included this as a brief point of discussion in the context of the suggested papers (lines 293-297).

Round 2

Reviewer 3 Report

Comments and Suggestions for Authors

Add an explanation of Figure1D to the caption, and change the caption location.